# A fear conditioned cue orchestrates a suite of behaviors in rats

Amanda Chu*†, Nicholas T Gordon, Aleah M DuBois, Christa B Michel, Katherine E Hanrahan, David C Williams, Stefano Anzellotti, Michael A McDannald*†

Department of Psychology and Neuroscience, Boston College, Chestnut Hill, United States

**Abstract** Pavlovian fear conditioning has been extensively used to study the behavioral and neural basis of defensive systems. In a typical procedure, a cue is paired with foot shock, and subsequent cue presentation elicits freezing, a behavior theoretically linked to predator detection. Studies have since shown a fear conditioned cue can elicit locomotion, a behavior that – in addition to jumping, and rearing – is theoretically linked to imminent or occurring predation. A criticism of studies observing fear conditioned cue-elicited locomotion is that responding is non-associative. We gave rats Pavlovian fear discrimination over a baseline of reward seeking. TTL-triggered cameras captured 5 behavior frames/s around cue presentation. Experiment 1 examined the emergence of danger-specific behaviors over fear acquisition. Experiment 2 examined the expression of danger-specific behaviors in fear extinction. In total, we scored 112,000 frames for nine discrete behavior categories. Temporal ethograms show that during acquisition, a fear conditioned cue suppresses reward seeking and elicits freezing, but also elicits locomotion, jumping, and rearing – all of which are maximal when foot shock is imminent. During extinction, a fear conditioned cue most prominently suppresses reward seeking, and elicits locomotion that is timed to shock delivery. The independent expression of these behaviors in both experiments reveals a fear conditioned cue to orchestrate a temporally organized suite of behaviors.

**\*For correspondence:**
amanda.chu@bc.edu (AC);
michael.mcdannald@bc.edu
(MAMcD)

†Lead contacts

**Competing interest:** The authors declare that no competing interests exist.

## Editor's evaluation

This is an important and timely characterization of a diversity of behaviors male and female rats exhibit during the acquisition of Pavlovian fear conditioning in a conditioned suppression procedure. The data are compelling and provide an exhaustive analysis of behavior in a complex associative learning paradigm that blends aversive Pavlovian and appetitive instrumental elements. The generalizability of these findings to other paradigms could be enhanced, however, with the inclusion of tests of cue responses in a neutral environment. These findings are likely to be of interest to those who study fear conditioning and associative learning more broadly in rodents.

## Introduction

Animals evolved defensive systems to detect and avoid predation. The predatory imminence continuum (PIC), a prominent theory of defensive behavior, identifies three defensive modes based on the proximity to predation: pre-encounter (leaving the safety of the nest), post-encounter (predator detected), and circa-strike (predation imminent or occurring) (*Fanselow and Lester, 1988*). Pavlovian fear conditioning has been extensively used to reveal the behavioral and neural underpinnings of defensive systems in rats (*Bolles and Collier, 1976*; *Fanselow, 1993*; *Killcross et al., 1997*; *McNally et al., 2011*). In a typical Pavlovian fear conditioning procedure, a rat is placed in a neutral context,

**eLife digest** Knowing that an animal is fearful is crucial for many psychology and neuroscience studies. For instance, this knowledge allows researchers to examine the brain pathways involved in processing and responding to fear.

Typically, researchers consider that a rodent is experiencing fear if it 'freezes' – a response which, in the wild, helps to evade detection by predators. In Pavlovian fear conditioning experiments, for example, rats and mice freeze when exposed to a stimulus (often a specific sound) previously associated with unpleasant sensations. However, rodents can also respond more actively to threats, for instance by running or jumping away. It remains unclear whether the 'fearful stimuli' used in Pavlovian approaches specifically elicits only freezing, or other fear-related behaviors as well.

To investigate this, Chu et al. used high-speed cameras to record rats' responses to a sound cue they had 'learned' to associate with a mild foot shock. In addition to freezing, the animals ran, jumped, stood on their hind legs and stopped their usual reward-seeking behavior in response to the cue. Crucially, these reactions were absent when the rats were exposed to sound cues not associated with pain.

Overall, these experiments demonstrate that Pavlovian conditioning can elicit a full range of fear-related behaviors beyond freezing. Understanding the neural activity behind these diverse responses could lead to more targeted therapies and interventions addressing the various ways stress and anxiety manifest in people.

and played an auditory cue whose termination coincides with foot shock delivery. Each PIC mode is characterized by a unique set of behaviors and, critically, each mode is thought to be captured by a unique epoch of a Pavlovian fear conditioning trial (*Fanselow et al., 2019*). The post-encounter mode is characterized by freezing, and is captured by cue presentation. Circa-strike is characterized by locomotion, jumping, and rearing, and is captured by shock delivery.

Freezing to a fear conditioned cue may be the most ubiquitous finding in all of behavioral neuroscience (*Blanchard and Blanchard, 1969*; *Bolles and Collier, 1976*; *Maren et al., 1997*; *Anagnostaras, 1999*; *Wilensky et al., 1999*; *Quirk, 2002*; *Koo et al., 2004*; *Rogers and Kesner, 2004*; *Iordanova et al., 2006*; *Shumake et al., 2014*; *Foilb et al., 2016*; *Furlong et al., 2016*). The relationship between freezing and Pavlovian fear conditioning is so strong that failing to observe freezing in defensive settings has been used to support assertions that Pavlovian fear conditioning did not occur (*Zambetti et al., 2021*). Cued fear as freezing has been further entrenched by historical observations that locomotion, jumping, and rearing (theorized circa-strike behaviors) are not elicited by fear conditioned cues (*Fanselow et al., 2019*). Instead, activity-promoting defensive behaviors are restricted to shock delivery (*Fanselow, 1982*) or to other sudden changes in stimuli (*Fadok et al., 2017*; *Totty et al., 2021*). Yet, locomotion, jumping, and rearing all readily occur in defensive settings (*Blanchard et al., 1986*; *Holland, 1979*; *Dielenberg and McGregor, 2001*). Most relevant, a fear conditioned cue can elicit locomotion, rapid forward movements termed 'darting' (*Gruene et al., 2015*; *Mitchell et al., 2022*).

The ability of a fear conditioned cue to elicit locomotion has been called into question (*Trott et al., 2022*). Trott et al. noted that in prior studies locomotion was greatest at cue onset – the time point most distal from shock delivery (*Gruene et al., 2015*; *Fadok et al., 2017*). Moreover, prior studies did not use associative controls (but see *Totty et al., 2021*) – essential to making claims that cue-elicited behaviors were due to a predictive relationship with foot shock. Using between-subjects designs in mice, Trott et al. ascribe the majority of cue-elicited locomotion to non-associative cue properties. The foundational study demonstrating the need for proper associative controls in *any* form of conditioning used Pavlovian fear conditioning (*Rescorla, 1967*). Not just all-or-none, the magnitude of a fear conditioned, cue-elicited response can scale with foot shock probability (*Rescorla, 1968*; *Ray et al., 2020*). *Rescorla, 1968* many foundational associative learning studies (*Kamin, 1969*; *Rescorla and Wagner, 1972*), relied on experiments that did not measure 'fear' with freezing, but with suppression of operant responding for reward (now termed conditioned suppression) (*Estes and Skinner, 1941*). Drawing from *Rescorla, 1968*, our laboratory has devised a robust, within-subjects Pavlovian fear conditioning procedure in which three cues predict unique foot shock probabilities: danger (p = 1),

uncertainty (p = 0.25), and safety (p = 0). Measuring conditioned suppression, we consistently observe complete behavioral discrimination: danger elicits greater suppression than safety, and uncertainty elicits suppression intermediate to danger and safety (*Wright et al., 2015*; *DiLeo et al., 2016*; *Walker et al., 2018*; *Ray et al., 2022*).

The goal of Experiment 1 was to construct comprehensive, temporal ethograms of rat behavior during discriminative Pavlovian fear conditioning, consisting of a danger, uncertainty, and safety cue. This would allow us to determine what behaviors come under the control of a fear conditioned cue, and how these behaviors are temporally organized. We had the ability to reveal freezing as the exclusive conditioned behavior, as prior studies have found positive relationships between conditioned freezing and conditioned suppression (*Bouton and Bolles, 1980*; *Mast et al., 1982*). Yet, we also had the ability to detect additional behaviors, as brain manipulations that impair conditioned freezing can have little or no impact on conditioned suppression (*McDannald, 2010*; *McDannald and Galarce, 2011*). A subgoal was to compare behaviors elicited by the deterministic danger cue, and the probabilistic uncertainty cue. The goal of Experiment 2 was to reveal which of these danger-elicited behaviors transferred to an extinction context in which shock and reward were not present. For Experiment 2, we simplified the discrimination procedure to include only the danger and safety cues.

Twenty-four rats (12 females; Experiment 1) and sixteen rats (8 females, Experiment 2) received Pavlovian fear discrimination. TTL-triggered GigE cameras were installed in behavioral boxes and programmed to capture frames at subsecond temporal resolution prior to and during cue presentation. 86,400 frames (Experiment 1) and 25,600 frames (Experiment 2) were hand scored for nine discrete behaviors reflecting reward (*Holland, 1977*), activity-suppressing fear (*Blanchard and Blanchard, 1969*; *Fanselow, 1982*), and activity-promoting fear (*Blanchard et al., 1986*; *Dielenberg and McGregor, 2001*; *Gruene et al., 2015*). Complete temporal ethograms were constructed during early, middle, and late conditioning sessions (Experiment 1), and for the two types of extinction tests (Experiment 2). Danger responding was compared to baseline and to safety, which served as an unpaired control cue. Behaviors elicited by the danger cue were considered associative (due to pairing with foot shock) if they differed both from baseline and from the safety cue. The temporal profile of responding was determined by tracking behavior change over cue presentation.

## Results

### Experiment 1

#### Conditioned suppression reveals complete discrimination

Twenty-four Long Evans rats (12 females) were trained to nose poke in a central port for food reward. Nose poking was reinforced on a 60-s variable interval schedule throughout behavioral testing. Independent of the poke-food contingency, auditory cues were played through overhead speakers, and foot shock delivered through the grid floor (*Figure 1A*). The experimental design consisted of three cues predicting unique foot shock probabilities: danger (p = 1), uncertainty (p = 0.25), and safety (p = 0) (*Figure 1B*). Behavior chambers were equipped with TTL-triggered cameras capturing 5 frames/s starting 5 s prior to cue presentation and continuing throughout the 10 s cue. TTL-triggered capture yielded 75 frames per trial, and 1200 frames per session. We aimed to capture 28,800 frames each session (1200 frames × 24 rats).

Our laboratory routinely observes complete behavioral discrimination between danger, uncertainty, and safety in female and male rats measuring conditioned suppression (*Walker et al., 2018*; *Wright et al., 2019*; *Ray et al., 2022*). Suppression ratios are calculated using baseline and cue nose poke rates: (baseline − cue)/(baseline + cue). Suppression ratios provide a continuous behavior measure, from no suppression (ratio = 0) to total suppression (ratio = 1). To determine if we observed complete behavioral discrimination in these 24 rats, we performed analysis of variance (ANOVA) for suppression ratios [factors: cue (danger vs. uncertainty vs. safety), session (17 total: 1 pre-exposure and 16 discrimination), and sex (female vs. male)]. Complete behavioral discrimination emerged over testing (*Figure 1C, D*). ANOVA found a significant main effect of cue and a significant cue × session interaction ($Fs > 6$, $ps < 0.0001$; see *Supplementary file 1* for specific values). Sex effects were apparent; ANOVA found a significant main effect of sex, as well as a significant cue × sex interaction and a cue × session × sex interaction ($Fs > 3$, $ps < 0.05$; *Supplementary file 1*). Female suppression ratios were

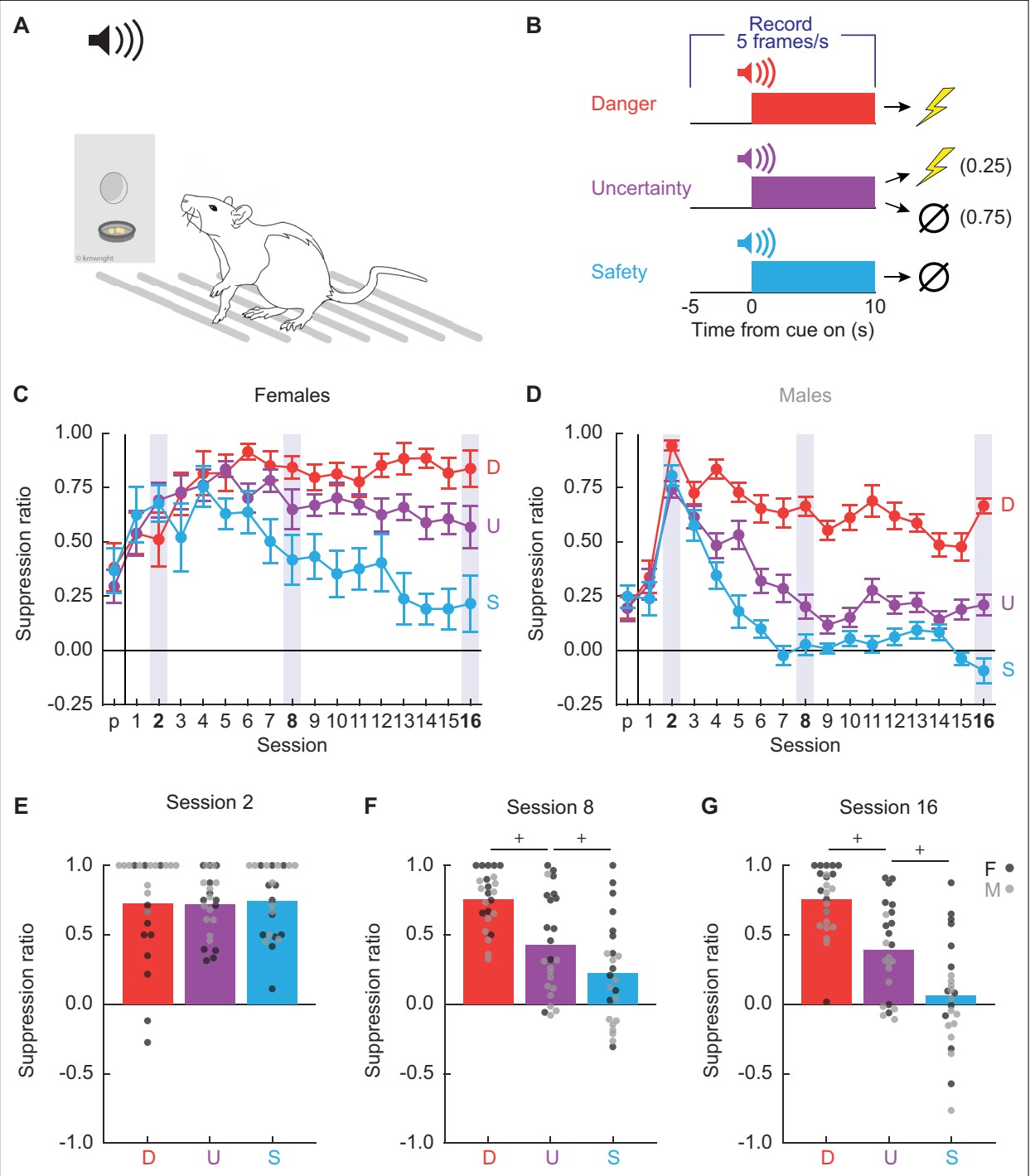

**Figure 1.** Experimental design and nose poke suppression. (**A**) Conditioned suppression procedure during which rats nose poke for food, while cues are played overhead and shocks delivered through floor. (**B**) Fear discrimination consisted of 10 s auditory cues predicting unique foot shock probabilities: danger (red; p = 1), uncertainty (purple; p = 0.25), and safety (blue; p = 0). Five video frames were captured per second, starting 5 s prior to cue onset and continuing through cue presentation. Mean ± standard error of the mean (SEM) suppression ratios for danger (red), uncertainty (purple), and safety (blue) from pre-exposure through discrimination session 16 are shown for (**C**) females and (**D**) males. Mean + individual suppression ratios for each cue are shown for (**E**) session 2, (**F**) session 8, and (**G**) session 16. Individuals represented by black (female) and gray (male) dots.+95% bootstrap confidence interval does not contain zero.

The online version of this article includes the following figure supplement(s) for figure 1:

**Figure supplement 1.** Body weight and baseline nose poke rate.

**Figure supplement 2.** Nose poke × discrimination.

higher to each cue across all discrimination sessions: danger ($t_{22}$ = 3.36, p = 0.003), uncertainty ($t_{22}$ = 7.14, p = 3.67 × 10$^{-7}$), and safety ($t_{22}$ = 4.40, p = 0.0002).

Sex differences in body weight and baseline nose poke rate existed prior to and throughout discrimination, with males weighing more and poking more than females (*Figure 1—figure supplement 1*). It is therefore possible that sex indirectly moderates conditioned suppression through effects on body weight or baseline nose poke rate. To determine this, we performed analysis of covariance (ANCOVA) for suppression ratios [factors: cue (danger vs. uncertainty vs. safety) and session (17 total: 1 pre-exposure and 16 discrimination)] using body weight or baseline nose poke rate as the covariate. ANCOVA with body weight found neither a significant body weight × cue interaction ($F_{(2,44)}$ = 2.97, p = 0.062) nor a significant body weight × cue × session interaction ($F_{32,704}$ = 1.40, p = 0.074). However, ANCOVA with baseline nose poke rate found a significant baseline × cue interaction ($F_{(2,44)}$ = 5.49, p = 0.007) but not a significant baseline × cue × session interaction ($F_{32,704}$ = 0.79, p = 0.79). Irrespective of sex, higher baseline nose poke rates predicted greater discrimination of danger and uncertainty (*Figure 1—figure supplement 2*).

Constructing behavioral ethograms for all 16 discrimination sessions would have required hand scoring 460,800 frames. To make scoring feasible and capture the emergence of discrimination, we selected sessions 2, 8, and 16. Suppression generalized to all cues during session 2 (*Figure 1E*). Behavioral discrimination emerged by session 8 (*Figure 1F*), and was at its most complete during session 16 (*Figure 1G*). Patterns were confirmed with 95% bootstrap confidence intervals (BCIs) which found no suppression ratio differences for any cue pair during session 2 (all 95% BCIs contained zero), but differences between all cue pairs during sessions 8 and 16 (no 95% BCIs contained zero).

Frames were hand scored for nine discrete behaviors: cup, freezing, grooming, jumping, locomotion, port, rearing, scaling, and stretching, plus 'background' (definitions in *Supplementary file 2*). Behavior categories and their definitions were based on prior work in appetitive conditioning (*Holland, 1977*), foot shock conditioning (*Fanselow, 1982*; *Blanchard et al., 1986*), as well as our own observations. Representative behavior frames are shown in *Figure 2*, *Videos 1–4* show example danger trials for four different rats (females in *Videos 1 and 3*, males in *Videos 2 and 4*).

## Temporal ethograms reveal shifting behavioral patterns over discrimination

The 86,400 scored frames allowed us to construct temporal ethograms for danger (*Figure 3A–C*), uncertainty (*Figure 3D–F*), and safety (*Figure 3G–I*) during sessions 2 (*Figure 3*, column 1), 8 (*Figure 3*, column 2), and 16 (*Figure 3*, column 3). Hand scoring showed high inter-rater reliability even when many behaviors were present in a single trial (*Figure 3—figure supplement 1*). Shifts in the composition of behavior from baseline to cue presentation were apparent across all ethograms. During session 2 (column 1), behavioral shifts lacked cue specificity. Temporal ethograms revealed danger, uncertainty, and safety to equally suppress grooming, port, and cup behavior, but increase freezing, and locomotion. Generalized cue control of behavior was supported by multiple analysis of variance (MANOVA) for all nine behavior categories [factors: cue (danger vs. uncertainty vs. safety), time (15 1 s bins: 5 s baseline → 10 s cue), and sex (female vs. male)] revealing a significant main effect of time ($F_{(126,2772)}$ = 2.37, p = 5.93 × 10$^{-15}$), but neither a significant main effect of cue ($F_{(18,74)}$ = 1.00, p = 0.47) nor a significant cue × time interaction ($F_{(252,5544)}$ = 1.12, p = 0.11). Cue-specific shifts in behavior were apparent by session 8 (column 2), and continued to session 16 (column 3). Now, MANOVA revealed significant main effects of cue (session 8, $F_{(18,74)}$ = 3.39, p = 0.0001; session 16, $F_{(18,74)}$ = 4.44, p = 0.000002), and significant cue × time interactions (session 8, $F_{(252,5544)}$ = 1.52, p = 3.31 × 10$^{-8}$; session 16, $F_{(252,5544)}$ = 1.52, p = 4.74 × 10$^{-7}$). Female-only ethograms are shown in *Figure 3—figure supplement 2*; male-only in *Figure 3—figure supplement 3*.

## Danger orchestrates a suite of behaviors

A central question driving this study is what behaviors come under the specific control of the fear conditioned, danger cue? To determine this, we focused on session 16, when discrimination was at its most complete. We first performed MANOVA for the 5 s baseline period [factors: cue (danger vs. uncertainty vs. safety), time (5, 1 s bins), and sex (female vs. male)]. As expected, MANOVA returned no main effect of cue, time, nor a cue × time interaction (*F*s < 1.5, ps > 0.1). Univariate ANOVA results were subjected to Bonferroni correction (p < 0.0055, 0.05/9 = 0.0055) to account for the nine separate analyses. Like for MANOVA, univariate ANOVA for each of the nine behaviors showed no main effect

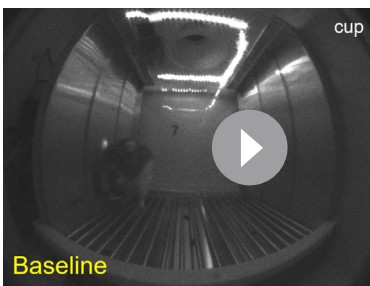

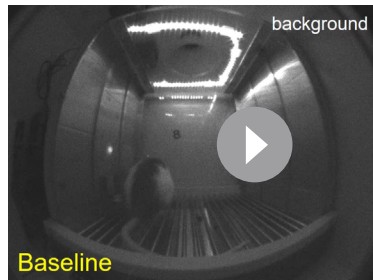

**Video 1.** Behavior during a single danger trial. Video shows the 75 sequential frames for a danger trial. Frames 1–25 are background and 26–75 are danger cue presentation. Observer judgment is shown in the top right for each frame. The specific trial is 23_16_12 (female rat 23, session 16, trial 12).
https://elifesciences.org/articles/82497/figures#video1

**Video 2.** Behavior during a single danger trial. Video shows the 75 sequential frames for a danger trial. Frames 1–25 are background and 26–75 are danger cue presentation. Observer judgment is shown in the top right for each frame. The specific trial is 24_16_16 (male rat 24, session 16, trial 16).
https://elifesciences.org/articles/82497/figures#video2

of cue, time, nor a cue × time interaction. In contrast to all other behaviors, univariate ANOVA for baseline freezing showed a main effect of sex ($F_{(1,22)}$ = 10.37, p = 0.004). ANOVA for freezing across the baseline and cue periods revealed a significant sex × cue × time interaction ($F_{(28,616)}$ = 1.94, p = 0.003). Females only froze during early danger presentation while males froze for the duration of danger presentation. The unique freezing pattern warrants separate consideration, which we return to later.

MANOVA was then performed for the 10 s cue period [factors: cue (danger vs. uncertainty vs. safety), time (10, 1 s bins), and sex (female vs. male)]. MANOVA returned significant main effects of cue and time, as well as a significant cue × time interaction ($F$s > 1.3, ps < 0.005). Of most interest, univariate ANOVA found a significant main effect of cue for six of the nine behaviors: port ($F_{(2,44)}$ = 32.15, p = 2.47 × 10⁻⁹, *Figure 4A*), cup ($F_{(2,44)}$ = 18.40, p = 0.00002, *Figure 4B*), locomote ($F_{(2,44)}$ = 6.33, p = 0.004, *Figure 4C*), jump ($F_{(2,44)}$ = 10.90, p = 0.0001, *Figure 4D*), rear ($F_{(2,44)}$ = 8.64, p = 0.001, *Figure 4E*), and freeze ($F_{(2,44)}$ = 13.86, p = 0.00002). Danger suppressed port and cup behavior (*Figure 4A, B*, line graphs), but promoted locomotion, jumping, and rearing (*Figure 4C–E*, line graphs). Danger-specific control of behavior was most apparent in the last 5 s of cue presentation (*Figure 4*, shaded region).

Claiming danger specificity requires that % behavior during the danger cue differs from baseline as well as the safety cue. To test this, we subtracted mean % behavior during the 5 s baseline from mean % behavior during the last 5 s of cue presentation, giving %Δ danger, %Δ uncertainty, and %Δ safety for each subject. We constructed 95% BCIs for each cue/behavior. 95% BCIs for %Δ danger did not contain zero for each of the five behaviors (*Figure 4*), meaning that levels of behavior during cue presentation differed from baseline. Danger presentation decreased port and cup behavior below

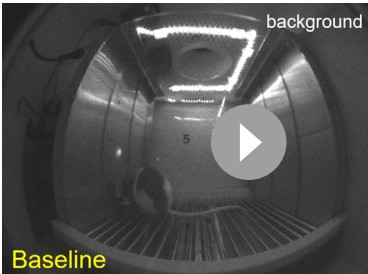

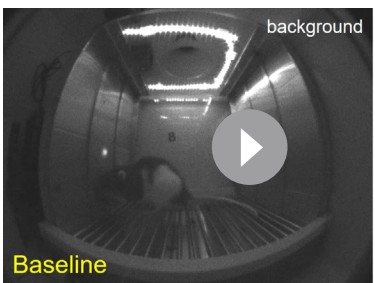

**Video 3.** Behavior during a single danger trial. Video shows the 75 sequential frames for a danger trial. Frames 1–25 are background and 26–75 are danger cue presentation. Observer judgment is shown in the top right for each frame. The specific trial is 5_16_11 (female rat 5, session 16, trial 11).
https://elifesciences.org/articles/82497/figures#video3

**Video 4.** Behavior during a single danger trial. Video shows the 75 sequential frames for a danger trial. Frames 1–25 are background and 26–75 are danger cue presentation. Observer judgment is shown in the top right for each frame. The specific trial is 4_16_3 (male rat 4, session 16, trial 3).
https://elifesciences.org/articles/82497/figures#video4

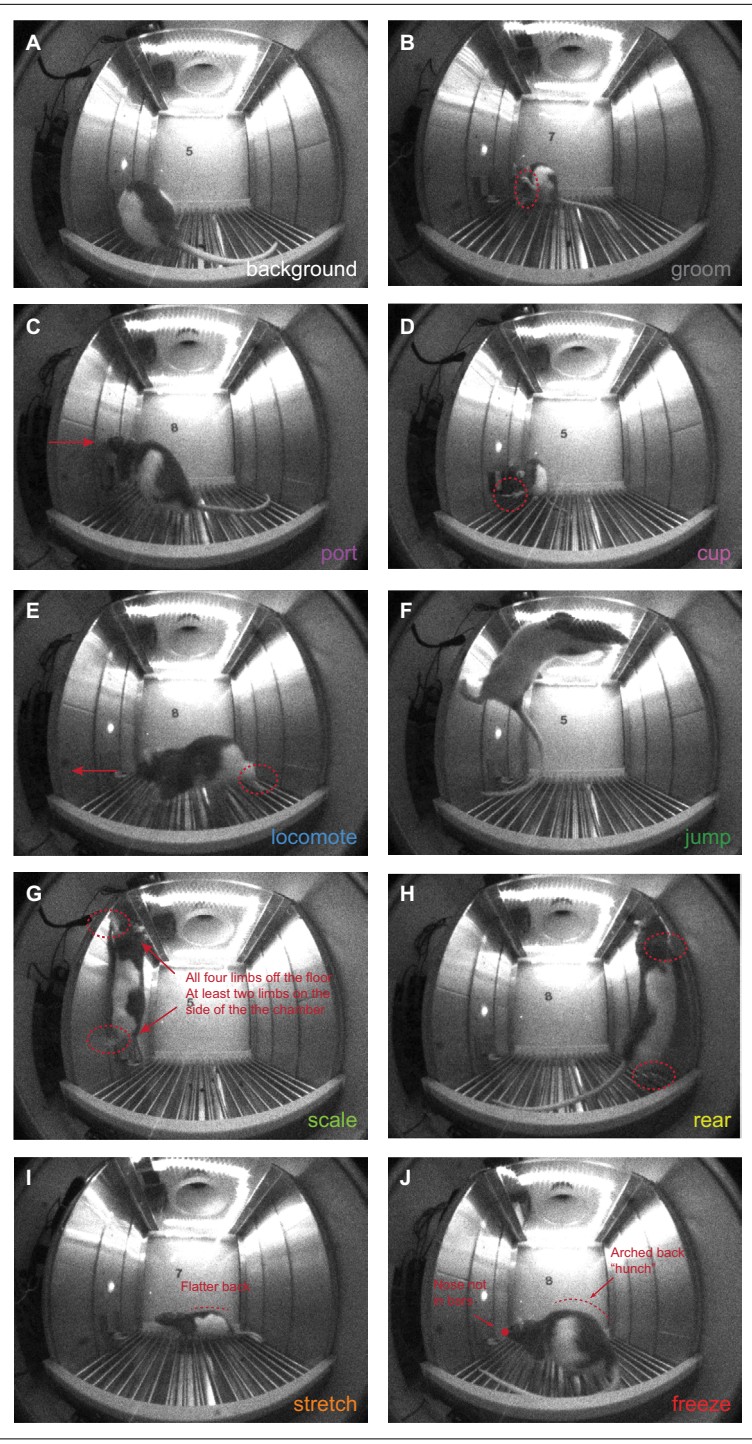

**Figure 2.** Representative behaviors. Representatives frames are shown for: (**A**) background, (**B**) groom, (**C**) port, (**D**) cup, (**E**) locomote, (**F**) jump, (**G**) scale, (**H**) rear, (**I**) stretch, and (**J**) freeze.

baseline, but increased locomotion, jumping, and rearing over baseline. 95% BCIs for %Δ uncertainty revealed increased locomotion and jumping, while 95% BCIs for %Δ safety revealed only decreased rearing. To demonstrate danger specificity, we subtracted %Δ safety from %Δ danger. We then constructed 95% BCIs for the difference score for each behavior. Confirming danger specificity (greater changes for danger than for safety), 95% BCIs did not contain zero for each of the five behaviors. Thus, danger specifically and selectively suppressed reward-related port and cup behavior, but promoted locomotion, jumping, and rearing.

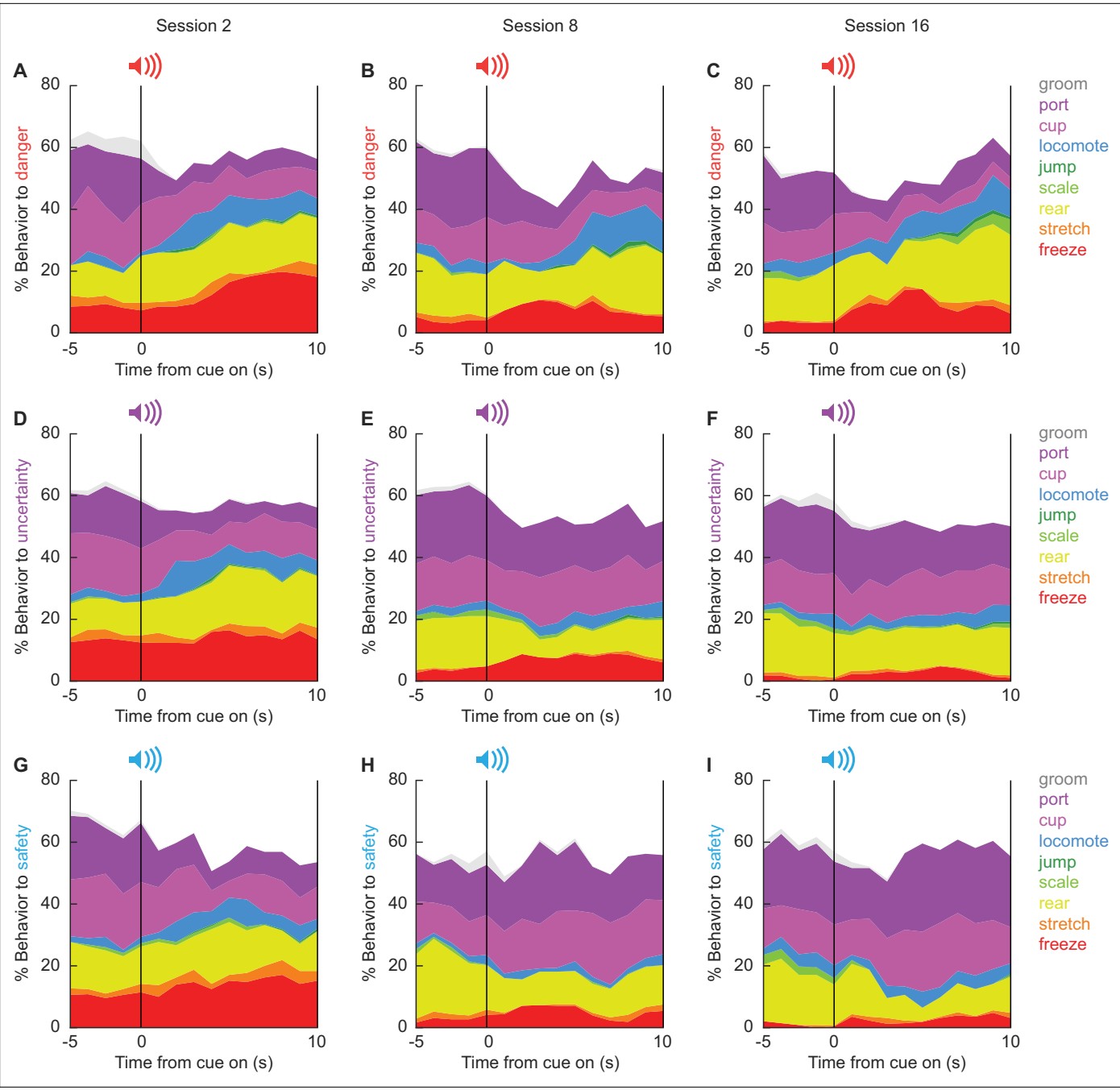

**Figure 3.** Temporal ethograms. Mean percent behavior from 5 s prior through 10 s cue presentation is shown for the danger cue during sessions (**A**) 2, (**B**) 8, and (**C**) 16; the uncertainty cue during sessions (**D**) 2, (**E**) 8, and (**F**) 16; and the safety cue during sessions (**G**) 2, (**H**) 8, and (**I**) 16. Behaviors are groom (gray), port (dark purple), cup (light purple), locomote (blue), jump (dark green), scale (light green), rear (yellow), stretch (orange), and freeze (red).

The online version of this article includes the following figure supplement(s) for figure 3:

**Figure supplement 1.** Inter-rater reliability.

**Figure supplement 2.** Female ethograms.

**Figure supplement 3.** Male ethograms.

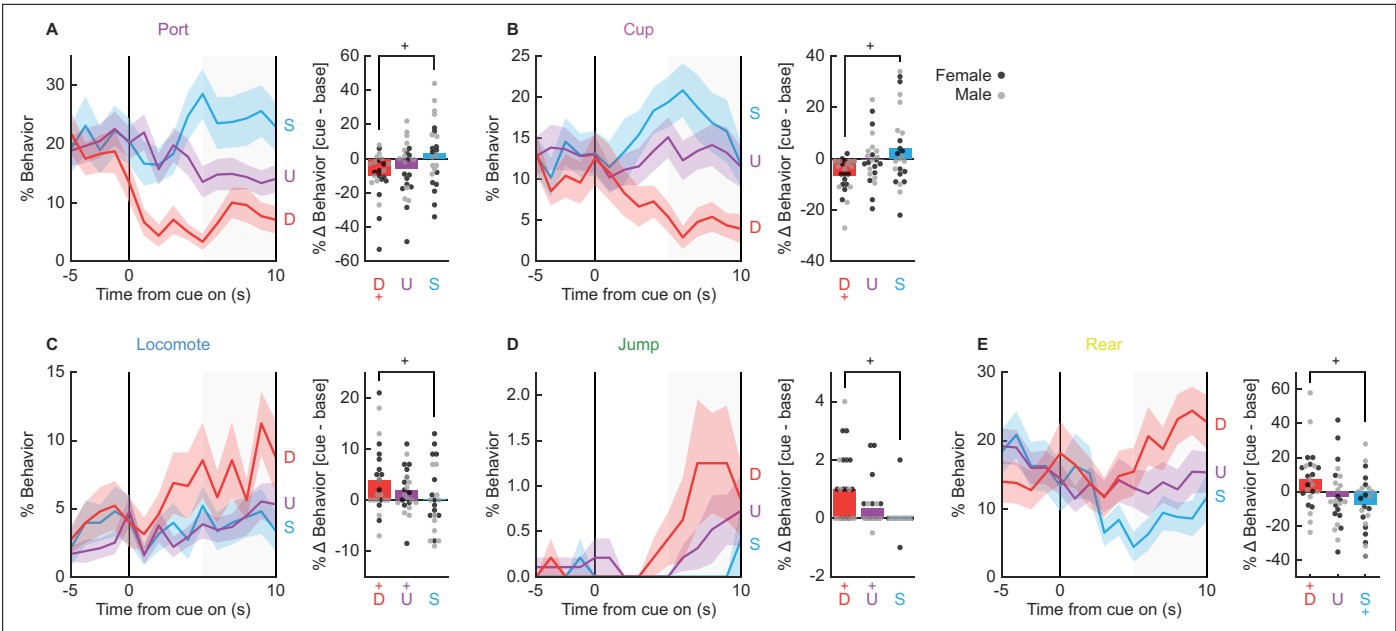

**Figure 4.** Danger-elicited behaviors. Line graphs show mean ± standard error of the mean (SEM) percent behavior from 5 s prior through 10 s cue presentation for danger (red), uncertainty (purple), and safety (blue) for (**A**) port, (**B**) cup, (**C**) locomote, (**D**) jumping, and (**E**) rearing. Bar plots show mean change in behavior from baseline (5 s prior to cue) compared to last 5 s of cue. Individuals represented by black (female) and gray (male) dots. +95% bootstrap confidence interval for danger vs. safety (black), danger vs. baseline (red), or safety vs. baseline (blue) comparison does not contain zero (black).

The online version of this article includes the following figure supplement(s) for figure 4:

**Figure supplement 1.** Comparison of session 16, danger-specific behaviors during sessions 2 and 16.

**Figure supplement 2.** Session 16, danger-specific behaviors during session 2.

## Associatively acquired behaviors generalize early

By the end of session 16 each rat had received 96 total foot shocks. It is possible that danger-specific control of multiple behaviors was only observed in session 16 because rats received far more cue–shock pairings than a typical Pavlovian conditioning procedure employs. Session 2 provided a comparison to numbers of cue–shock pairings more typical of fear conditioning studies; rats had received 12 total foot shocks by session's end. The key question was whether pattern of danger-elicited behaviors in session 2 resembled the pattern in session 16, or if a fundamentally different pattern was observed. To determine this, we performed univariate ANOVA for danger [factors: session (2 vs. 16) and time (15, 1 s bins)] for each of the five behaviors showing session 16 selectivity (*Figure 4—figure supplement 1*). Confirming near identical temporal patterns of behavior expression during sessions 2 and 16, ANOVA found no significant session × time interaction for any behavior [port ($F_{(14,322)}$ = 0.45, p = 0.96), cup ($F_{(14,322)}$ = 0.61, p = 0.86), locomote ($F_{(14,322)}$ = 1.09, p = 0.37), jump ($F_{(14,322)}$ = 1.23, p = 0.25), and rear ($F_{(14,322)}$ = 0.92, p = 0.54)]. Thus, danger orchestrated a suite of behaviors even early in discrimination. Recall that early discrimination (session 2) was marked by non-specific cue control of behaviors. This would mean that associatively acquired behaviors initially generalized to uncertainty and safety – and that discrimination consisted of restricting behavior to danger. In support, univariate ANOVA for session 2 [factors: cue (danger vs. uncertainty vs. safety), time (15, 1 s bins), and sex (female vs. male)] found no cue × time interaction for any of the five, danger-specific behaviors (all *F*s < 1.2, all ps > 0.3).

## Sex informs the temporal pattern of freezing

We return to the case of freezing; the most measured overt fear conditioned behavior. We again focus on session 16 during which discrimination was most complete. Female and male rats differed in the temporal pattern and cue specificity of freezing. Females showed higher baseline freezing levels, a rapid increase in freezing that was specific to danger in the first 5 s, then became non-specific and declined back to baseline levels in the last 5 s (*Figure 5A*). In contrast, males show little

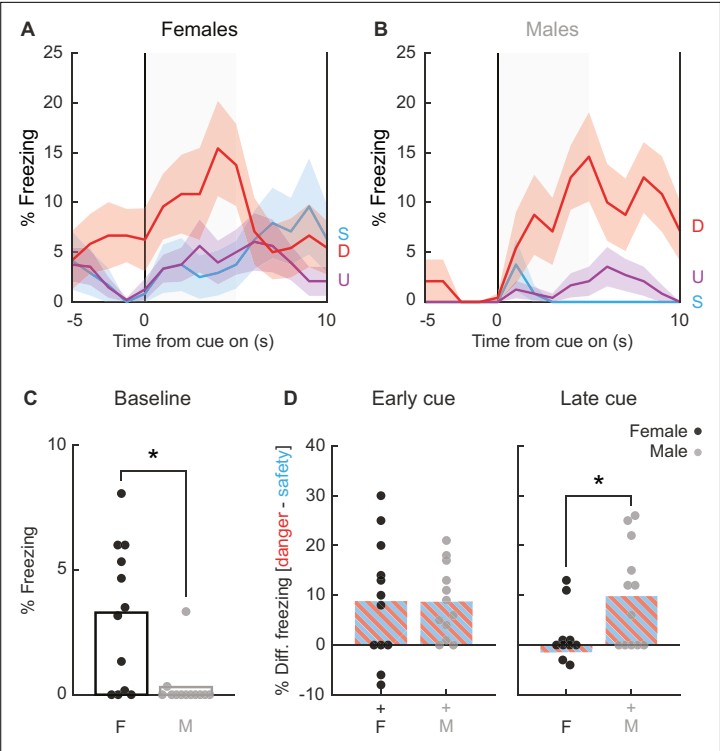

**Figure 5.** Special case of freezing. Line graphs show mean ± standard error of the mean (SEM) percent freezing from 5 s prior through 10 s cue presentation for danger (red), uncertainty (purple), and safety (blue) for (**A**) females and (**B**) males. (**C**) Percent freezing during baseline (5 s prior to cue) is shown for females (black) and males (gray). (**D**) Mean differential freezing to danger and safety is shown for females (black, left) and males (gray, right) during early cue (first 5 s of cue, left) and late cue (last 5 s of cue, right). Mean ± SEM percent freezing change from baseline (5 s prior to cue) compared to last 5 s of danger (red), uncertainty (purple), and safety (blue) for (E) females and (F) males.

The online version of this article includes the following figure supplement(s) for figure 5:

**Figure supplement 1.** Session 2 freezing.

baseline freezing and danger-specific freezing increases that persisted throughout cue presentation (*Figure 5B*). Baseline freezing differences were confirmed with independent samples *t*-test ($t_{22}$ = 3.22, p = 0.0039; *Figure 5C*). Confirming sex differences in the temporal pattern of freezing, differential freezing to danger and safety was equivalent in females and males during early cue presentation ($t_{22}$ = 0.02, p = 0.98; *Figure 5D*, left), but differed during late cue ($t_{22}$ = 2.80, p = 0.01; *Figure 5D*, right). Generalized freezing to all cues was observed during session 2, with freezing increases more evident in males (*Figure 5—figure supplement 1*). Thus, discrimination consisted of restricting freezing to danger in males, and selectively freezing to early danger presentation in females.

## Danger-elicited behaviors are independently expressed

Danger suppression of reward-related port and cup behaviors could simply be the byproduct of danger-elicited freezing. Such a relationship has previously been reported (*Bouton and Bolles, 1980*; *Mast et al., 1982*). To examine the relationship between reward-related behaviors and freezing, in addition to other possible behavior–behavior relationships, we calculated %Δ behavior for early (first 5 s) and late (last 5 s) danger presentation for the six danger-elicited behaviors: cup, port, locomote, jump, rear, and freeze. We constructed 12 × 12 matrices containing the *R* values (*Figure 6A*) and p values (*Figure 6B*) for the Pearson's correlation coefficient for each behavior–behavior comparison during the two danger periods. Surprisingly, only one behavior–behavior relationship was observed during the early danger presentation period (*Figure 6A*, upper left quadrant). Early rearing and early cup behavior were negatively correlated (R = −0.43, p = 0.03, but note this would not survive Bonferroni correction). Even more, no behavior–behavior relationships were observed during late danger

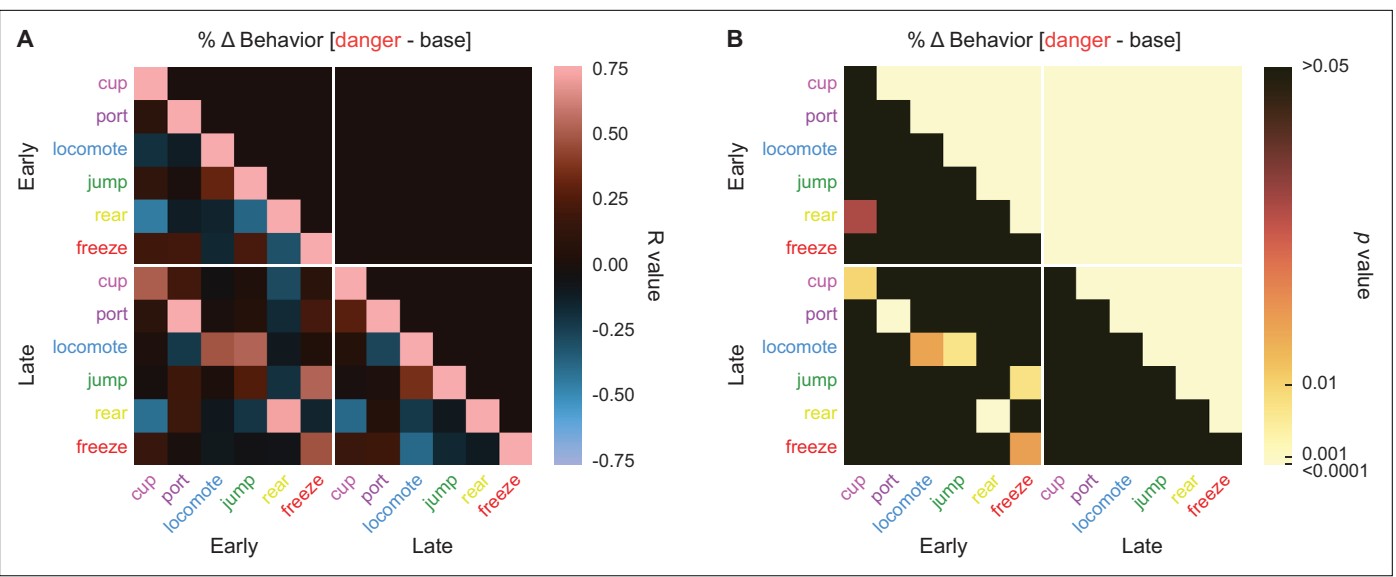

**Figure 6.** Behavior–behavior correlations. (**A**) A correlation matrix for the six cue-specific behaviors port (dark purple), cup (light purple), locomote (blue), jump (dark green), rear (yellow), and freeze (red) comparing mean percent behavior during early (first 5 s) and late (last 5 s) cue is shown. Lighter red values indicate positive *R* values, lighter blue values indicate negative *R* values. Black indicates *R* = 0. p values associated with each associated *R* value are shown in (**B**). Black indicates p values greater than 0.05, while increasingly lighter values indicate lower p values.

presentation (*Figure 6A*, lower right quadrant). These results suggest the six behaviors are more or less expressed independently of one another.

Maybe our analysis cannot detect behavior–behavior relationships? To test this, we compared behaviors across the early and late danger periods. Now, the correlation matrix revealed a band of positive *R* values cutting diagonally across the bottom left quadrant. Five of the 6 behaviors showed positive early–late relationships with themselves: cup ($R = 0.51$, $p = 0.01$), port ($R = 0.87$, $p = 2.67 \times 10^{-8}$), locomote ($R = 0.48$, $p = 0.017$), rear ($R = 0.71$, $p = 7.92 \times 10^{-5}$), and freeze ($R = 0.48$, $p = 0.017$). In other words, changes in cup behavior evident during early danger presentation persisted to late danger presentation. Jumping was an exception to this trend, as there was no relationship between early and late jumping levels to danger. Overall, danger-elicited behaviors were expressed independently of one another.

## Experiment 2

In Experiment 2, we aimed to answer two questions: (1) were the danger-elicited behaviors during discrimination in Experiment 1 dependent on foot shock delivery, and (2) were these behaviors due to the presence of the reward apparatus? To answer this, rats received danger vs. safety discrimination, then were given extinction tests with reward apparatus absent or present. During extinction testing, we captured and hand scored behavior frames before, during, and following cue presentation.

### Conditioned suppression reveals complete discrimination during extinction with reward apparatus present

Sixteen Long Evans rats (8 females) were trained to nose poke in a central port for food reward as in Experiment 1. Nose poking was reinforced on a 60-s variable interval schedule throughout behavioral testing. Independent of the poke-food contingency, auditory cues were played through overhead speakers, and foot shock delivered through the grid floor (*Figure 7A*). The experimental design consisted of two cues deterministically predicting foot shock: danger ($p = 1$) and safety ($p = 0$) (*Figure 7A*).

To determine if we observed complete behavioral discrimination, we performed ANOVA for suppression ratios [factors: cue (danger vs. safety), session (13 total: 1 pre-exposure and 12 discrimination), and sex (female vs. male)]. Complete behavioral discrimination emerged over testing (*Figure 7B*). ANOVA found a significant main effect of cue and a significant cue × session interaction ($Fs > 8$, $ps <$

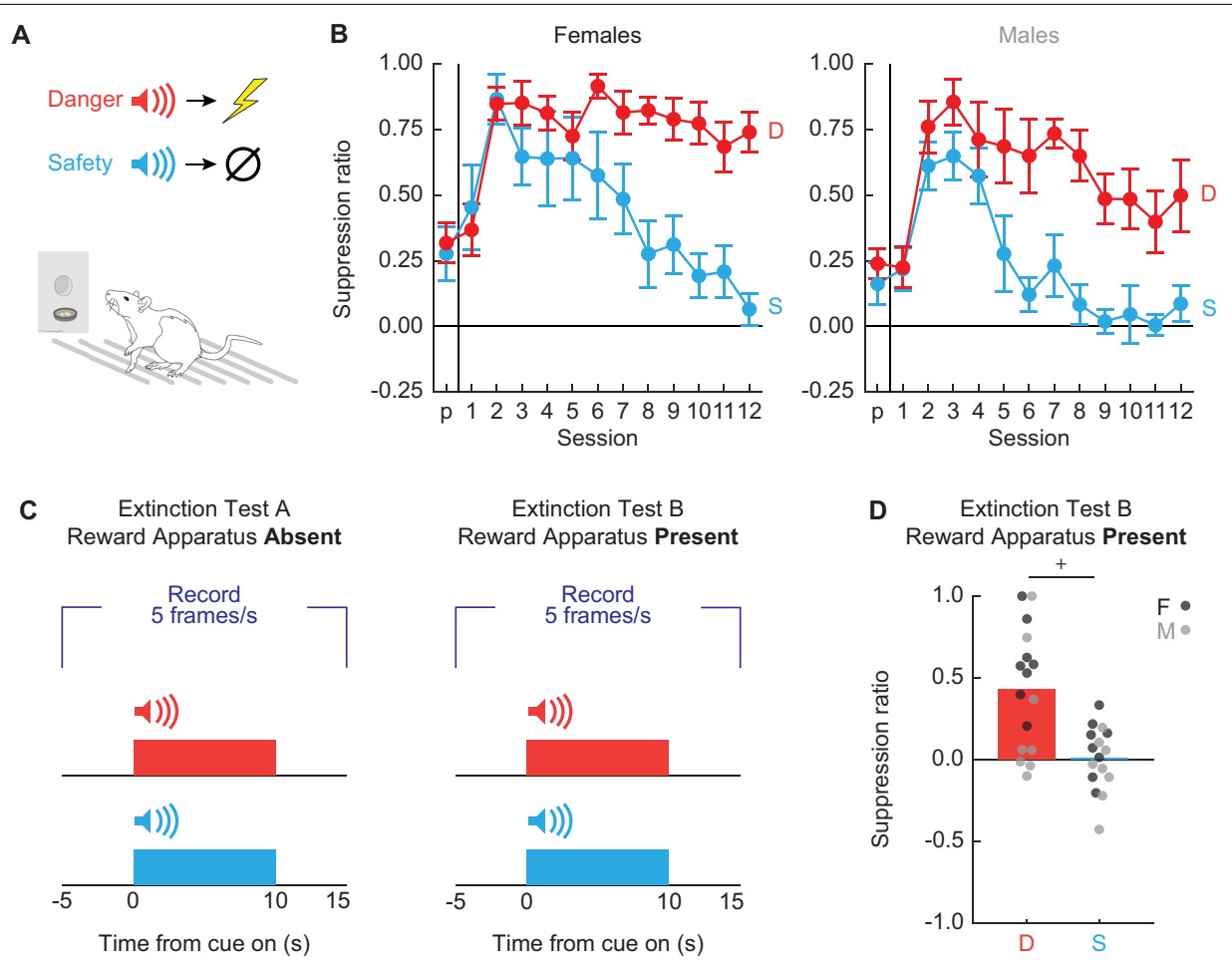

**Figure 7.** Experimental design and nose poke suppression. (**A**) Conditioned suppression procedure during which rats nose poke for food, while danger (red; p = 1) and safety (blue; p = 0) cues are played overhead and shocks delivered through floor. (**B**) Mean ± standard error of the mean (SEM) suppression ratios for danger (red) and safety (blue) from pre-exposure through discrimination session 12 are shown for (left) females and (right) males. (**C**) Rats received one extinction test with reward apparatus absent (left), and another with reward apparatus present (right), counterbalanced. Five video frames were captured per second, starting 5 s prior to cue onset and continuing through 5 s after cue offset. (**D**) Mean + individual suppression ratios for each cue are shown for extinction with reward apparatus present. Individuals represented by black (female) and gray (male) dots. +95% bootstrap confidence interval does not contain zero.

0.0001; see *Supplementary file 3* for specific values). No significant main effect or interactions with sex were observed. Following the 12 discrimination sessions, each rat received two extinction test sessions. In both test sessions each cue was presented four times. In one test session, the nose poke and food cup were absent while in the other test session the nose poke and food cup were present (*Figure 7C*). Test order was fully counterbalanced. 95% BCIs for differential suppression ratio (danger − safety) during extinction test with the reward present revealed complete discrimination (*Figure 7D*). The 95% BCI did not contain zero [lower bound = 0.24, upper bound = 0.60].

We captured 25,600 total frames (800 frames/test × 16 rats × 2 tests) during extinction testing. Frames were hand scored for nine discrete behaviors: cup, freezing, grooming, jumping, locomotion, port, rearing, scaling, and stretching, plus 'background' as in Experiment 1, with the exception that if a trial did not have the reward apparatus present, then food cup and nose poke were not scored.

### Danger-elicited locomotion peaks when foot shock would have occurred

The 25,600 scored frames allowed us to construct temporal ethograms for danger (*Figure 8A, B*) and safety (*Figure 8C, D*), during the extinction test with reward apparatus absent (*Figure 8*, column 1), and during the extinction test with the reward apparatus present (*Figure 8*, column 2). Hand scoring

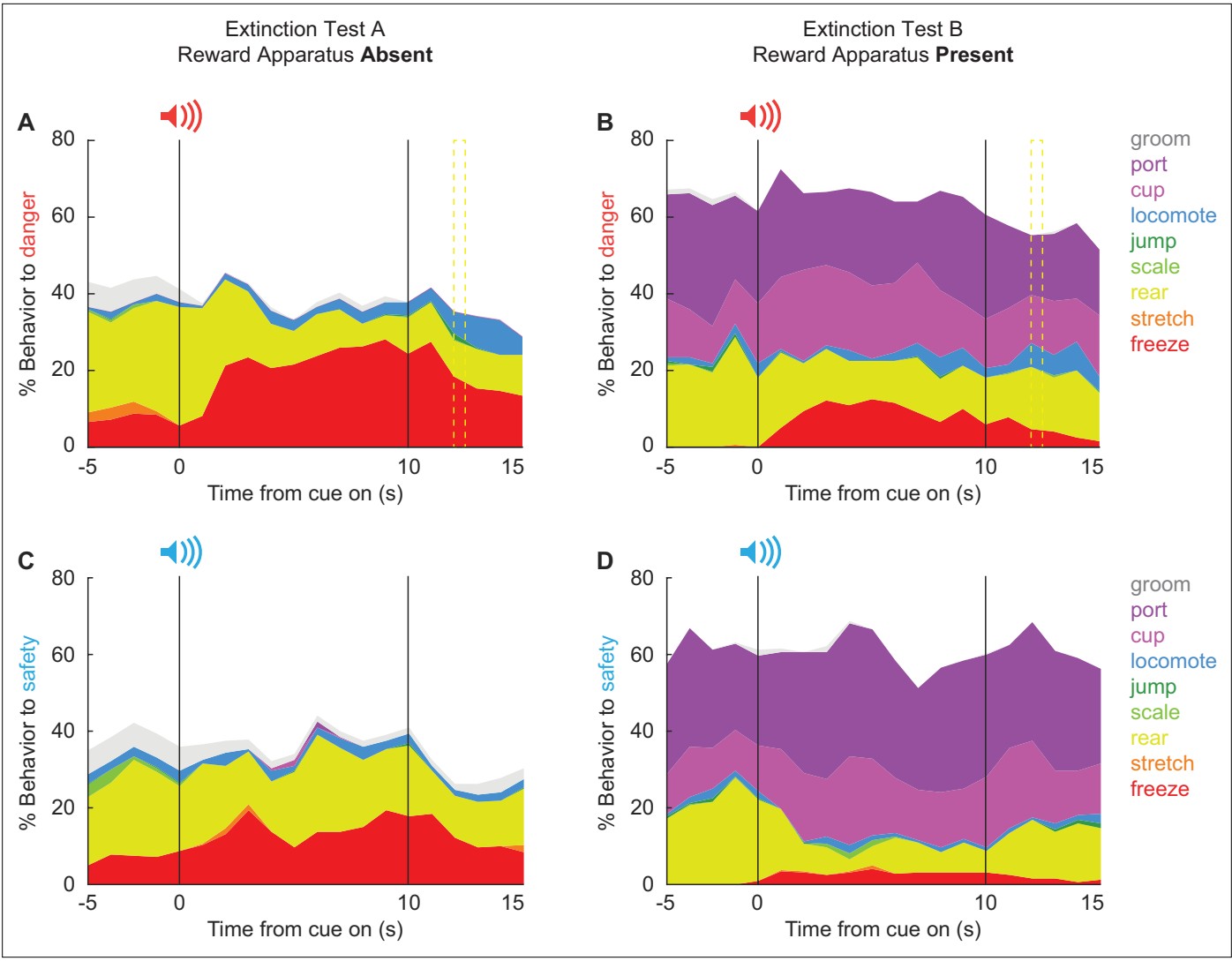

**Figure 8.** Temporal ethograms during extinction. Mean percent behavior from 5 s prior through 5 s following cue offset is shown for the danger cue during extinction with (**A**) reward apparatus absent and (**B**) reward apparatus present; and the safety cue during extinction with (**C**) reward apparatus absent and (**D**) reward apparatus present. Behaviors are groom (gray), port (dark purple), cup (light purple), locomote (blue), jump (dark green), scale (light green), rear (yellow), stretch (orange), and freeze (red). Note, port and cup are not shown for A and C because the food cup and nose port were absent.

The online version of this article includes the following figure supplement(s) for figure 8:

**Figure supplement 1.** Inter-rater reliability.

showed high inter-rater reliability even when many behaviors were present in a single trial (*Figure 8— figure supplement 1*). Cue-specific changes in behavior during and following cue presentation were evident. In support, MANOVA [factors: sex (female vs. male), test type (absent vs. present), order (absent first vs. present first), cue (danger vs. safety), and time (20 1 s bins: 5 s baseline → 10 s cue → 5 s post cue)] for the seven behaviors common to both tests [groom, locomote, jump, scale, rear, stretch, and freeze] revealed a significant main effect of time ($F_{(133,1596)}$ = 2.14, p = 9.44 × 10⁻¹²) and a significant cue × time interaction ($F_{(133,1596)}$ = 1.46, p = 0.001).

Of the seven behaviors, danger only increased locomotion during both test types (*Figure 9A, B*). In support, univariate ANOVA for locomotion [Bonferroni-corrected p < 0.007 (0.05/7 = 0.007); factors: sex (female vs. male), test type (absent vs. present), order (absent first vs. present first), cue (danger vs. safety), and time (20 1 s bins: 5 s baseline → 10 s cue → 5 s post cue)] found a significant cue × time interaction ($F_{(19,228)}$ = 3.12, p = 0.000026). Danger-elicited locomotion was most prominent following cue offset, around the time shock would have occurred. 95% BCIs revealed danger-elicited

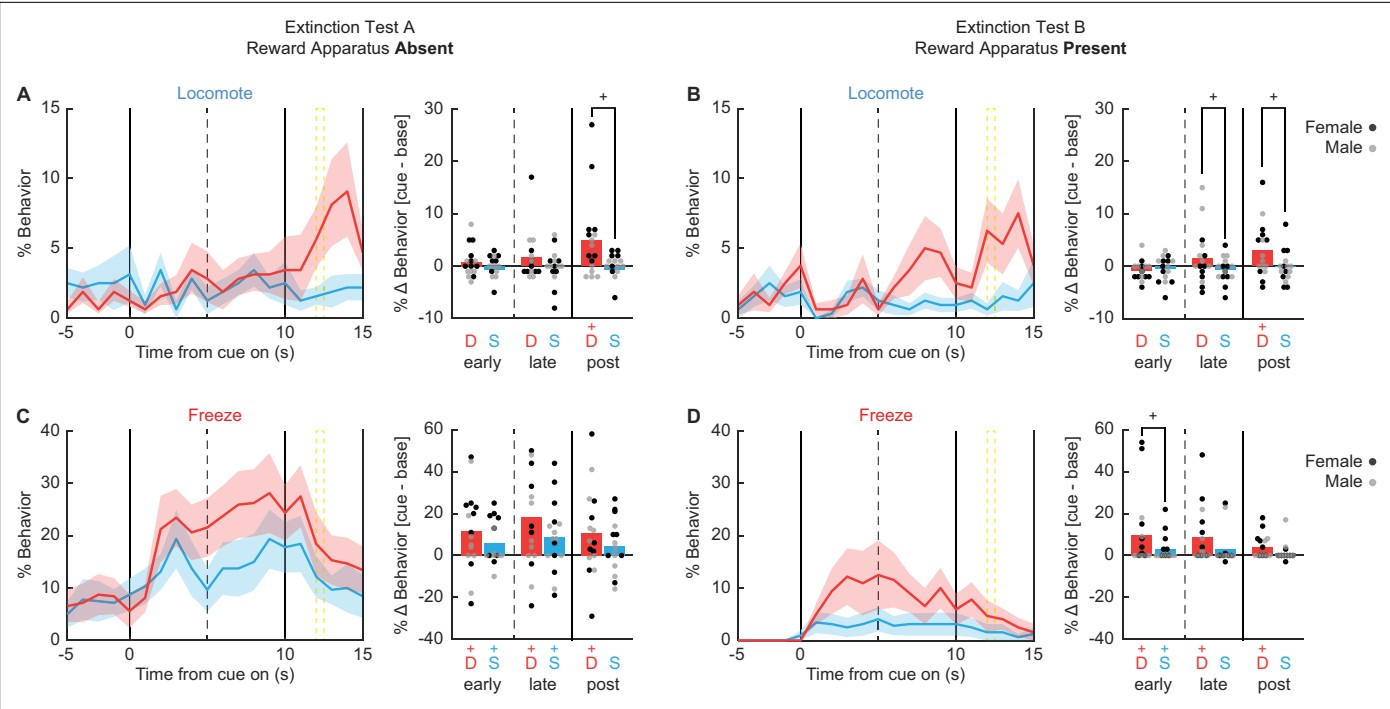

**Figure 9.** Danger elicits locomotion. Line graphs show mean ± standard error of the mean (SEM) percent behavior from 5 s prior through 10 s cue presentation for danger (red) and safety (blue) for locomotion during the (**A**) reward apparatus absent extinction test and (**B**) reward apparatus present extinction test. Bar plots show mean change in behavior from baseline (5 s prior to cue) compared to early (first 5 s), late (last 5 s), and post (5 s after offset) cue periods. Individuals represented by black (female) and gray (male) dots. The same is shown for freezing (**C, D**). +95% bootstrap confidence interval for danger vs. safety (black), danger vs. baseline (red), or safety vs. baseline (blue) comparison does not contain zero (black).

The online version of this article includes the following figure supplement(s) for figure 9:

**Figure supplement 1.** Female behavior during extinction testing.

**Figure supplement 2.** Male behavior during extinction testing.

**Figure supplement 3.** Freezing separated by test order.

locomotion to exceed baseline and safety cue levels during the 5 s post-cue periods for both the Absent (*Figure 9A*) and Present tests (*Figure 9B*). Additionally, the 95% BCI revealed danger-elicited locomotion to exceed safety-elicited locomotion during the late cue period during the Present test, though danger-elicited locomotion did not exceed baseline (*Figure 9B*). Locomotion never increased during safety trials (all 95% BCIs contain zero). Danger-elicited locomotion occurred regardless of test order, as ANOVA revealed no significant order interactions ($Fs < 1.5$, $ps > 0.2$). Sex partially mediated the temporal expression of locomotion, with ANOVA finding a significant sex × cue × time interaction ($F_{(19,228)} = 2.34$, $p = 0.002$). Females showed more robust post-cue, danger locomotion during both test types (*Figure 9—figure supplement 1*). Males showed more robust danger-elicited locomotion during the late cue period during the Present test (*Figure 9—figure supplement 2*). The results reveal that danger-elicited locomotion transfers to extinction settings when both foot shock and the reward apparatus were absent.

## Freezing is less dangerspecific and is sensitive to time, test type, and order

Unlike locomotion, there was lesser evidence of danger-specific freezing during extinction testing (*Figure 9C, D*). Most notably, univariate ANOVA [correction and factors identical to locomotion] found that the cue × time interaction failed to achieve significance ($F_{(19,228)} = 1.25$, $p = 0.011$). When organizing % freezing by test type, there was no period (early cue, late cue, and post cue) during which freezing increases over baseline were selective to danger (*Figure 9C, D*). The only period during which freezing to danger exceeded freezing to safety was the early cue period when the reward apparatus was present (*Figure 9D*, right). Though even during this period increases in freezing to safety were observed. Instead, freezing tended to generalize to safety; meaning it was cue evoked but

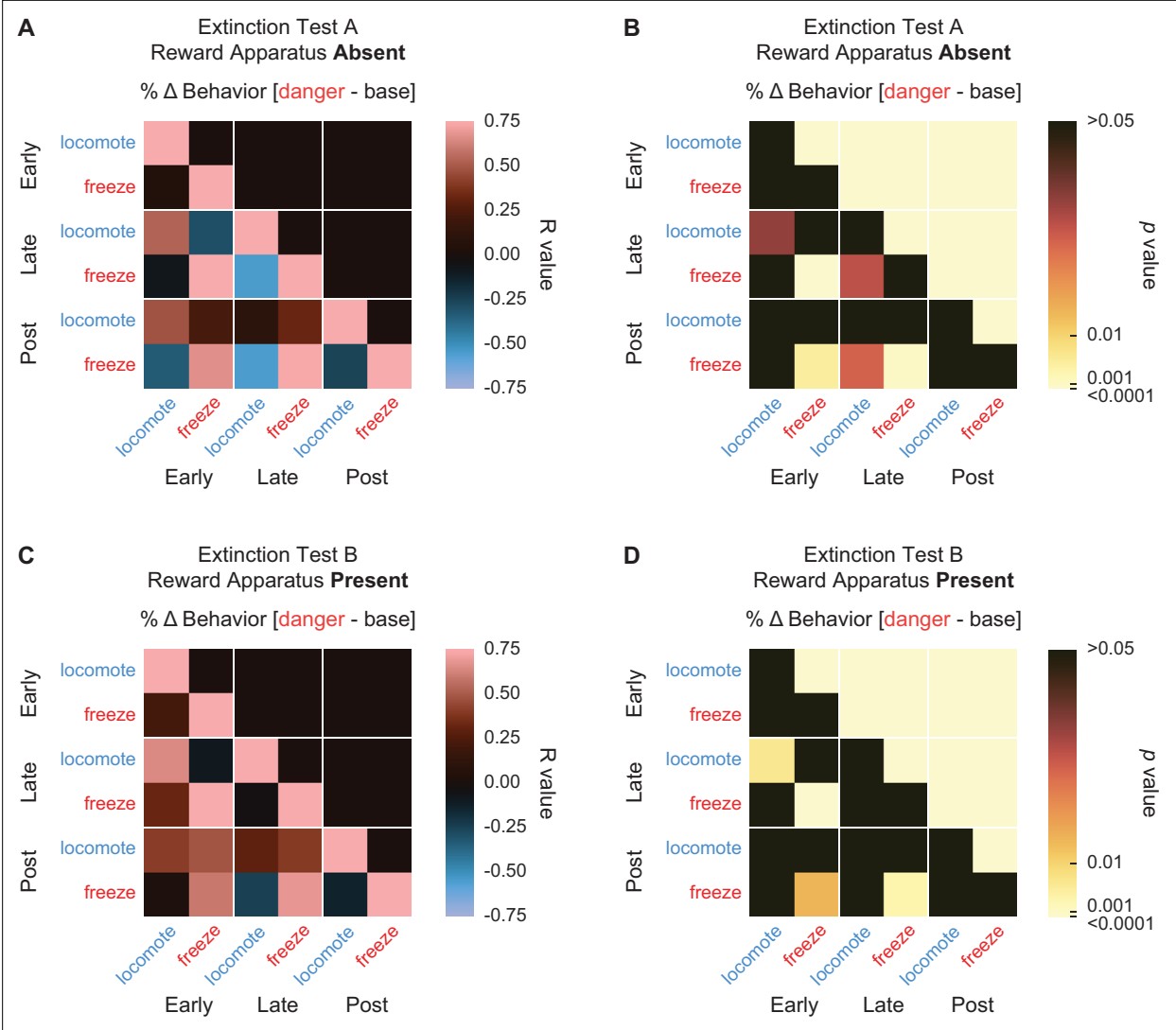

**Figure 10.** Behavior–behavior correlations during extinction. (**A**) A correlation matrix for locomote (blue) and freeze (red) comparing mean percent behavior during early (first 5 s), late (last 5 s), and post (5 s after) cue period is shown for the reward absent extinction test. Lighter red values indicate positive *R* values, lighter blue values indicate negative *R* values. Black indicates *R* = 0. p values associated with each associated *R* value are shown in (**B**). Black indicates p values greater than 0.05, while increasingly lighter values indicate lower p values. Same shown for behavior correlations during reward present extinction test (**C, D**).

not cue specific. Additionally, freezing was more prominent during extinction testing with the reward apparatus absent. In support, univariate ANOVA revealed significant main effects of time ($F_{(19,228)}$ = 5.13, p = 3.64 × 10$^{-10}$) and test ($F_{(1,12)}$ = 21.20, p = 0.001). Like freezing, neither rearing nor jumping showed evidence of danger specificity with univariate ANOVA for each finding no significant cue × time interaction ($F$s < 1.5, ps > 0.2).

However, order mediated the specificity of danger-elicited freezing. Rats receiving the Present test first showed selective and differential freezing to danger (*Figure 9*, *Figure 9—figure supplement 3A*). Rats receiving the Absent test first showed no evidence of selective and differential freezing to danger (*Figure 9*, *Figure 9—figure supplement 3B*). In support, univariate ANOVA returned a significant order × cue × time interaction ($F_{(19,228)}$ = 2.14, p = 0.002). Of note, no significant order × cue × time interaction was observed for locomotion ($F_{(19,228)}$ = 1.03, p = 0.43). The same rats that showed robust danger-elicited locomotion across both test types showed danger-elicited freezing that was sensitive to test order.

## Danger-elicited behaviors are independently expressed during extinction

We were interested to see if there were relationships between danger-elicited locomotion and freezing during extinction testing. To determine this we calculated Pearson's correlation coefficients (*R* value) for individual freezing and locomotion (% behavior over baseline) during early, late, and post-danger cue periods in extinction sessions with reward apparatus absent (*Figure 10A, B*) and with reward apparatus present (*Figure 10C, D*). As in Experiment 1, we found no evidence of inhibitory relationships between locomotion and freezing. That is, no comparison found a negative *R* value. This was true both within and between trial periods. Instead, and like for Experiment 1, correlational analyses reveal significant, positive relationships within behaviors across trial periods. These positive relationships were more prominent during extinction testing with the reward apparatus present. In particular, freezing was positively correlated across all trial periods during the present extinction sessions [early–late $R = 0.82$, p = $1.05 \times 10^{-4}$; early–post $R = 0.60$, p = 0.015; post–late $R = 0.68$, p = 0.0036]. These results demonstrate that opposing danger-elicited behaviors are independently expressed during extinction.

## Discussion

We set out to quantify behaviors elicited by a fear conditioned, danger cue. Consistent with virtually all studies of Pavlovian fear conditioning (but see *Amorapanth et al., 1999*), we observed danger-elicited freezing over the course of acquisition (Experiment 1) and during extinction testing (Experiment 2). Yet, freezing was not the dominant danger-elicited behavior. Instead, danger orchestrated a suite of behaviors. During Experiment 1, danger suppressed reward behavior directed toward the site of food delivery and the location of the rewarded action. Even more, danger-elicited locomotion, jumping, and rearing. During Experiment 2, danger again suppressed rewarded action and continued to elicit locomotion. During both experiments, freezing was most prominent at danger cue onset. Locomotion was most prominent toward danger cue offset (Experiment 1), and when shock would have occurred in extinction (Experiment 2).

Before discussing our results further, an important limitation should be raised. 40–50% of frames could not be assigned to a specific behavior and were labeled as background. This was due to three main factors. First, in order to objectively hand score many behaviors, we developed mutually exclusive, specific definitions. Our strict definitions meant erring on the side of labeling a behavior background if there was uncertainty in judgment. Second, use of a single, side view camera meant the observer could not view a rat's forelimbs or face when the rat was turned away from the camera. If forelimb and face position could not be determined the frame was labeled background. Finally, transition behaviors (e.g., switching from rearing to locomotion) and other behaviors (e.g., sniffing and turning) that did not fit into one of the nine behavior definitions were labeled background. The upside of this limitation is high confidence in behavior judgments and high inter-rater reliability for those judgments.

Studies assessing auditory fear conditioning in a neutral context routinely report freezing to account for >80% of behavior during cue presentation (*Bolles and Collier, 1976*; *Maren et al., 1997*; *Anagnostaras, 1999*; *Wilensky et al., 1999*; *Quirk, 2002*; *Koo et al., 2004*; *Rogers and Kesner, 2004*; *Iordanova et al., 2006*; *Shumake et al., 2014*; *Foilb et al., 2016*; *Furlong et al., 2016*). The sheer number of demonstrations, and number of groups independently observing dominant freezing, puts us firmly in the minority. Placing us further in the minority, we observe danger-elicited locomotion, jumping, and rearing. These behaviors are characterized by lateral and vertical movements, polar opposites to the immobility that characterizes freezing. A commonality of the studies above was use of contexts in which only cues and shocks were delivered, with shocks being more intense than shocks used in our studies. These experimental settings favor freezing. It is likely that our inclusion of competing, reward behaviors and use of lower shock intensities permitted a broader range of danger-elicited behaviors to be observed (*Holland, 1979*; *Mitchell et al., 2022*).

Indeed, we are not the first group to observe locomotion, jumping, or rearing in defensive settings in rats. Using more traditional Pavlovian fear conditioning designs, Shansky and colleagues have observed 'darting', rapid forward movements across the test chamber, to a fear conditioned cue (*Gruene et al., 2015*). While more readily observed in female rats, darting can be observed in males under certain experimental conditions (*Mitchell et al., 2022*). Our definition of locomotion aligns

well with the Shansky lab definition of darting. We found that danger-elicited locomotion was equally apparent during extinction, and more robust than freezing. While locomotion was observed across all rats, female locomotion was better timed to shock delivery. Our results join previous studies in demonstrating a fear conditioned cue promotes movement in rats (*Bolles and Collier, 1976*; *Totty et al., 2021*).

Jumping is elicited in rats by hypoxia (decreasing oxygen levels in the air) – a life-threatening condition (*Spiacci et al., 2015*). More relevant to our study, the Blanchards observed jumping in defensive settings in rats (*Blanchard et al., 1986*). In their procedure, a rat was placed at the end of an inescapable hallway, then a human experimenter slowly approached. Rats initially froze when the experimenter was distant (1–2 m away), but switched from freezing to jumping as the experimenter drew near (<1 m). Our observation of danger-elicited jumping during fear acquisition, and its preferential expression at the end of danger presentation, mirrored the defensive jumping pattern observed in the Blanchard's study.

Rearing (*Lever et al., 2006*) is elicited by visual cues predicting moderate foot shock. *Holland, 1979* found that a mix of rearing and freezing are acquired to a visual cue paired with low intensity foot shock. A visual cue paired with intense foot shock exclusively produces freezing. The foot shock intensity we used in both experiments (0.5 mA) is more consistent with the low intensities in the *Holland, 1979* experiment. *Dielenberg and McGregor, 2001* found that rats exposed to a recently worn cat collar, with an opposing box to hide in, show 'vigilant rearing' to the cat collar (*Dielenberg and McGregor, 2001*). Rearing was never observed in a control condition. While we cannot claim vigilance, we find that danger promotes rearing during fear acquisition.

Temporal ethograms revealed that during fear acquisition, jumping and rearing were most prevalent at the end of cue presentation – when foot shock was imminent. This was in contrast to freezing which was prominent during early danger presentation for both females and males, but only shown by males at the end of cue presentation. Though unlike Experiment 1, in which foot shock was present, we found no evidence of danger-elicited jumping and rearing during fear extinction. Because jumping and rearing are vertical behaviors, they may be avoidant or escape responses. The rat is trying to leave the floor before the shock comes on. This interpretation is supported by the finding that these responses peaked just before shock presentation in Experiment 1. Removing foot shock in Experiment 2 may have removed the impetus for avoidance and escape. However, it could be equally likely that these behaviors are more sensitive context change. Experiment 2 also found that freezing transferred less well to the extinction context in which reward was absent.

Our findings accord well with the PIC theory of defensive behavior (*Fanselow and Lester, 1988*). PIC theory identifies three defensive modes: pre-encounter (e.g., leaving the safety of the nest to forage), post-encounter (predator detected), and circa-strike (predation inevitable or occurring). Analogs to PIC modes are identified in a Pavlovian fear conditioning trial (*Fanselow et al., 2019*). Pre-encounter mode may correspond to leaving the home cage and being placed in the experimental chamber where foot shocks occur. Post-encounter mode corresponds to presentation of the fear conditioned cue. Circa-strike mode is said to correspond to foot shock delivery. It is argued that circa-strike behaviors (locomotion, jumping, and rearing) are not observed toward the end of danger presentation because rats do not time shock delivery. In support, extending cue duration in traditional cued fear conditioning paradigms does not result in shifts from freezing to locomotion, jumping, and rearing toward cue offset (*Fanselow et al., 2019*).

Here, we find expected patterns of defensive behavior in unexpected epochs of Pavlovian conditioning trials. Early danger freezing by all rats (females and males) gives way to a late mix of danger-elicited behaviors that included locomotion, jumping, and rearing (Experiment 1) or late locomotion (Experiment 2). Why do we observe late danger control of circa-strike behaviors? Hunger and the availability of a rewarded action may provide an impetus for shock timing. Timing would allow rats to minimize the display of defensive behaviors and maximize reward seeking. In support, presenting long duration danger cues in a conditioned suppression setting results in timing of fear responding. With experience, rats show little suppression of reward seeking to danger onset, which ramps toward shock delivery (*Rosas and Alonso, 1996*). Supporting the minimization of defensive behavior in reward settings, foot shocks signaled by danger will strongly suppress reward seeking only early in fear conditioning. Shock-induced suppression quickly wanes and with experience, shock delivery will paradoxically facilitate reward seeking (*LaBarbera and Caul, 1976*; *Strickland*

*et al., 2021*). Shock timing information is readily apparent in the ventrolateral periaqueductal gray, a brain region central to defensive behavior (*Fanselow, 1993*; *Carrive et al., 1997*; *Mobbs et al., 2007*; *McDannald, 2010*; *Tovote et al., 2016*; *Arico et al., 2017*). Populations of ventrolateral periaqueductal gray neurons show little responding upon danger presentation, but ramp firing toward shock delivery (*Ozawa et al., 2017*; *Wright and McDannald, 2019*; *Wright et al., 2019*). Our results support the PIC theory of defensive behavior but demonstrate that the relationship between defensive mode and Pavlovian conditioning trial epoch is not fixed, but depends on the experimental setting.

A secondary goal of Experiment 1 was to compare defensive behaviors elicited by a deterministic, danger cue and a probabilistic, uncertainty cue. In our setting, uncertainty is not simply a diminished version of danger. Indeed, uncertainty only promoted a subset of danger-elicited behaviors: locomotion and jumping. Most surprising was the inability of uncertainty to suppress reward behaviors directed toward the food cup and port. This is particularly puzzling because using suppression ratios, we found uncertainty to produce robust suppression of nose poking. What is going on here? Food cup, port, and poke behavior lie on a rewarded action continuum. Food cup means the rat is in the area of food delivery – but is most distal from the rewarded action. Port means the rat is around or in the site of the rewarded action, but only poke requires the rat to be fully engaged in performing the rewarded action (nose all the way into the port). Danger suppresses all reward behavior regardless of proximity to rewarded action. In contrast, uncertainty selectively suppresses reward behavior most proximal to the rewarded action.

By comprehensively quantifying behavior and constructing temporal ethograms, we found a fear conditioned cue to independently control at least six distinct behaviors during fear acquisition and three distinct behaviors during fear extinction. Though our study was exclusively behavioral, we feel our results have implications for investigations into the neural basis of fear learning and the organization of defensive behavior. Most important is that a fear conditioned cue does not simply elicit freezing. Behaviors elicited by a fear conditioned cue are the product of many factors: species, sex, age, behavioral setting, and experimenter-determined parameters (CS/US type, duration, and intensity; trial number, inter-trial interval [ITI], and more). In our view, freezing is a common – not dominant – defensive behavior because the field has favored behavioral settings and experimenter-determined parameters that maximize the expression of 'fear' through freezing (*McDannald, 2023*). Here, we show that a relatively simple modification of the rat's behavioral setting – access to a rewarded action – is sufficient to de-emphasize freezing and promote the expression of many additional behaviors. Most prominent of these is locomotion. Even more, Pavlovian fear conditioning over a baseline of reward seeking reveals a temporally organized sequence of cue-elicited defensive behaviors predicted by PIC theory. The independent expression of these behaviors is appealing for studies attempting to link discrete behavioral sequelae of 'fear' to distinct neural circuits, breathing new life into a classic Pavlovian fear conditioning procedure (*Estes and Skinner, 1941*).

## Materials and methods

All procedures were performed in the Boston College Animal Care Facility in accordance with NIH and Boston College guidelines. The Boston College experimental protocol supporting these procedures is 2024-001.

## Subjects

For Experiment 1, 24 adult Long Evans rats (12 female) weighing 196–298 g arrived from Charles River Laboratories on postnatal day 55. Rats were single-housed on a 12-hr light cycle (lights off at 6:00 pm) and maintained at their initial body weight with standard laboratory chow (18% Protein Rodent Diet #2018, Harlan Teklad Global Diets, Madison, WI). Water was available ad libitum in the home cage. For Experiment 2, sixteen adult Long Evans rats (eight females) were housed and maintained as described above. All protocols were approved by the Boston College Animal Care and Use Committee and all experiments were carried out in accordance with the NIH guidelines regarding the care and use of rats for experimental procedures.

## Behavior apparatus

The apparatus for Pavlovian fear discrimination consisted of four individual chambers with aluminum front and back walls, clear acrylic sides and top, and a grid floor. LED strips emitting 940 nm light were affixed to the acrylic top to illuminate the behavioral chamber for frame capture. 940-nm illumination was chosen because rats do not detect light wavelengths exceeding 930 nm (*Nikbakht and Diamond, 2021*). Each grid floor bar was electrically connected to an aversive shock generator (Med Associates, St. Albans, VT). An external food cup, and a central port equipped with infrared photocells were present on one wall. Auditory stimuli were generated with an Arduino-based device and presented through two speakers mounted on the ceiling.

## Pellet exposure and nose poke shaping

Rats were food restricted and specifically fed to maintain their body weight throughout behavioral testing. Each rat was given 4 g of experimental pellets in their home cage in order to overcome neophobia. Next, the central port was removed from the experimental chamber, and rats received a 30-min session in which one pellet was delivered every minute. The central port was returned to the experimental chamber for the remainder of behavioral testing. Each rat was then shaped to nose poke in the central port for experimental pellet delivery using a fixed ratio schedule in which one nose poke into the port yielded one pellet. Shaping sessions lasted 30 min or until approximately 50 nose pokes were completed. Each rat then received six sessions during which nose pokes into the port were reinforced on a variable interval schedule. Session 1 used a variable interval 30-s schedule (poking into the port was reinforced every 30 s on average). All remaining sessions used a variable interval 60-s schedule. For the remainder of behavioral testing, nose pokes were reinforced on a variable interval 60-s schedule independent of cue and shock presentation.

## Cue pre-exposure

Each rat was pre-exposed to the three cues to be used in Pavlovian discrimination in one session. Auditory cues consisted of repeating motifs of broadband click, phaser, or trumpet. This 37-min session consisted of four presentations of each cue (12 total presentations) with a mean ITI of 2.5 min. Trial type order was randomly determined by the behavioral program and differed for each rat, each session.

## Pavlovian fear discrimination

### Experiment 1

Each rat received 16, 48-min sessions of fear discrimination. Each session consisted of 16 trials, with a mean ITI of 2.5 min. Auditory cues were 10 s in duration. Each cue was associated with a unique foot shock probability (0.5 mA, 0.5 s): danger, p = 1.00; uncertainty, p = 0.25; and safety, p = 0.00. Foot shock was administered 2 s following the termination of the auditory cue on danger and uncertainty-shock trials. Auditory identity was counterbalanced across rats. Each session consisted of four danger trials, two uncertainty-shock trials, six uncertainty-omission trials, and four safety trials. Trial type order was randomly determined by the behavioral program and differed for each rat, each session.

### Experiment 2

Each rat received 12, 48-min sessions of fear discrimination. Each session consisted of eight trials, with a mean ITI of 3.5 min. Auditory cues were 10 s in duration. Each cue was associated with a unique foot shock probability (0.5 mA, 0.5 s): danger, p = 1.00 and safety, p = 0.00. Foot shock was administered 2 s following the termination of the auditory cue on danger trials. Auditory identity was counterbalanced across rats. Each session consisted of four danger trials and four safety trials. Trial type order was randomly determined by the behavioral program and differed for each rat, each session.

## Fear extinction

For Experiment 2, each rat received two types of extinction test: one with the port and food cup present and one with the port and food cup absent. Test type order was counterbalanced across rats with half receiving the port and cup present first. Extinction sessions were 48 min in duration, and consisted of four danger and four safety trials, with a mean ITI of 3.5 min. Auditory cues were 10 s

in duration. Foot shocks were not delivered. Auditory identity of danger and safety were maintained from discrimination, which was counterbalanced. Trial type order was randomly determined by the behavioral program and differed for each rat.

## Calculating suppression ratio

Time stamps for cue presentations, shock delivery, and nose pokes (photobeam break) were automatically recorded by the Med Associates program. Baseline nose poke rate was calculated for each trial by counting the number of pokes during the 20 s pre-cue period and multiplying by 3. Cue nose poke rate was calculated for each trial by counting the number of pokes during the 10 s cue period and multiplying by 6. Nose poke suppression was calculated as a ratio: (baseline poke rate – cue poke rate)/(baseline poke rate + cue poke rate). A suppression ratio of '1' indicated complete suppression of nose poking during cue presentation relative to baseline. A suppression ratio of indicated '0' indicates equivalent nose poke rates during baseline and cue presentation. Gradations in suppression ratio between 1 and 0 indicated intermediate levels of nose poke suppression during cue presentation relative to baseline. Negative suppression ratios indicated increased nose poke rates during cue presentation relative to baseline.

## Frame capture system

Behavior frames were captured using Imaging Source monochrome cameras (DMK 37BUX28; USB 3.1, 1/2.9″ Sony Pregius IMX287, global shutter, resolution 720 × 540, trigger in, digital out, C/CS-mount). Frame capture was triggered by the Med Associates behavior program. The 28 V Med Associates pulse was converted to a 5-V TTL pulse via Adapter (SG-231, Med Associates, St. Albans, VT). The TTL adapter was wired to the camera's trigger input. Captured frames were saved to a PC (OptiPlex 7470 All-in-One) running IC Capture software (Imaging Source). For Experiment 1, frame capture began precisely 5 s before cue onset and continued throughout 10 s cue presentation. Frames were captured at a rate of 5 per second, with a target of capturing 75 frames per trial (5 frames/s × 15 s = 75 frames), and 1200 frames per session (75 frames/trial × 16 trials = 1200 frames). For Experiment 2, frame capture began 5 s before cue onset and continued throughout 10 s cue presentation and 5 s after cue termination. Frames were captured at a rate of 5 per second, with a target of capturing 100 frames per trial (5 frame/s × 20 s = 100 frames), and 800 frames per session (100 frames/trial × 8 trials = 800 frames).

## Post-acquisition frame processing

### Experiment 1

We aimed to capture 1200 frames per session, and selected sessions 2, 8, and 16 for hand scoring. A Matlab script sorted the 1200 frames into 16 folders, one for each trial, each containing 75 frames. Each 75-frame trial was made into a 75-slide PowerPoint presentation to be used for hand scoring.

### Experiment 2

We aimed to capture 800 frames per session, and selected extinction sessions 1 and 2 for hand scoring. A Matlab script sorted the 800 frames into 8 folders, one for each trial, each containing 100 frames. Each 100-frame trial was made into a 100-slide PowerPoint presentation to be used for hand scoring.

## Anonymizing trial information

For Experiment 1, a total of 1152 trials of behavior were scored from the 24 rats over the 3 sessions of discrimination (16 trials per session). For Experiment 2, a total of 256 trials were scored from 16 rats over the 2 extinction sessions (8 trials per session). We anonymized trial information in order to score behavior without bias. The numerical information from each trial (session #, rat #, and trial #) was encrypted as a unique number sequence. A unique word was then added to the front of this sequence. The result was that each of the trials was converted into a unique word + number sequence. For example, trial ac01_02_07 (rat #1, session #2, and trial #7) would be encrypted as: abundant28515581. The trials from Experiment 1 were randomly assigned to five observers. 256 trials from Experiment 2 were randomly assigned to seven observers. The result of trial anonymization was that observers were completely blind to subject, trial type, and session number. Furthermore,

random assignment meant that the 16 or 8 trials composing a single session were scored by different observers.

## Behavior categories and definitions

Frames were scored as one of ten mutually exclusive behavior categories, defined as follows:

### Background

Specific behavior cannot be discerned because the rat is turned away from the camera or position of forepaws is not clear, or because the rat is not engaged in any of the other behaviors.

### Cup

Any part of the nose above the food cup but below the nose port.

### Freeze

Arched back and stiff, rigid posture in the absence of movement, all four limbs on the floor (often accompanied by hyperventilation and piloerection). Side-to-side head movements and up and down head movements that do not disturb rigid posture are permitted. Activity such as sniffing or investigation of the bars is not freezing. Freezing, as opposed to pausing, is likely to be 3 or more frames (600+ms) long.

### Groom

Any scratching, licking, or washing of the body.

### Jump

All four limbs off the floor. Includes hanging which is distinguished when hind legs are hanging freely.

### Locomote

Propelling body across chamber on all four feet, as defined by movement of back feet. Movement of back feet with front feet off the floor is rearing.

### Port

Any part of the nose in the port. Often standing still in front of the port but sometimes tilting head sideways with the body off to the side of the port.

### Rear

One or two hind legs on the grid floor with both forepaws off the grid floor and not on the food cup. Usually (not always) stretching to full extent, forepaws usually (not always) on top of side walls of the chamber, often pawing walls; may be accompanied by sniffing or slow side-to-side movement of head. Does not include grooming movements or eating, even if performed while standing on hind legs.

### Scale

All four limbs off the floor but at least two limbs on the side of the chamber. Standing on the food cup counts as scaling.

### Stretch

Body is elongated with the back posture 'flatter' than normal. Stretching is often accompanied by immobility, like freezing, but is distinguished by the shape of the back.

## Frame scoring system

Frames were scored using a specific procedure. Frames were first watched in real time in Microsoft PowerPoint by setting the slide duration and transition to 0.19 s, then playing as a slideshow. Behaviors clearly observed were noted. Next, the observer went through all the frames scoring one behavior at a time. A standard scoring sequence was used: port, cup, rear, scale, jump, groom, freeze,

locomote, and stretch. When the specific behavior was observed in a frame, that frame was labeled. Once all behaviors had been scored, the video was re-watched for freezing. The unlabeled frames were then labeled 'background'. Finally, all background frames were checked to ensure they did not contain a defined behavior.

## Inter-observer reliability

### Experiment 1
To assess inter-observer reliability, we selected 12 trials from outside sessions 2, 8, and 16, six from females and six from males. Each of our five observers scored these 12 trials, interweaving the 12 comparison trials with the primary data trials. As a result, each observer scored 900 comparison frames which were then used to assess inter-observer reliability.

### Experiment 2
Inter-observer reliability was assessed as described in Experiment 1. Eight trials from outside extinction sessions 1 and 2 were selected for comparison. Each observer scored 800 comparison frames which were then used to assess inter-observer reliability.

## Statistical analyses
ANOVA was performed for body weight, baseline nose poke rate, suppression ratios, and specific behaviors. Sex was used as a factor for all analyses. Cue, session, and time were used as factors when relevant. Univariate ANOVA following MANOVA used a Bonferroni-corrected p value significance of 0.0055 (0.05/9) to account for the nine quantified behaviors. MANOVA was performed for the nine quantified behaviors with factors of sex, cue, and time. Pearson's correlation coefficient was used to examine the relationship between baseline nose poke rate and body weight, baseline nose poke rate and cue discrimination, as well as the relationship between danger cue-elicited behaviors during early and late cue presentation in session. Within-subject comparisons were made using 95% BCIs with the Matlab bootci function. Comparisons were said to differ when the 95% BCI did not contain zero. Between subject's comparisons were made using independent samples $t$-test.

---

# Additional information

### Funding

| Funder | Grant reference number | Author |
| --- | --- | --- |
| National Institutes of Health | MH117791 | Michael A McDannald |

The funders had no role in study design, data collection, and interpretation, or the decision to submit the work for publication.

### Author contributions
Amanda Chu, Conceptualization, Data curation, Software, Formal analysis, Supervision, Funding acquisition, Validation, Visualization, Methodology, Writing – original draft, Project administration, Writing – review and editing; Nicholas T Gordon, Aleah M DuBois, Christa B Michel, Katherine E Hanrahan, David C Williams, Data curation, Formal analysis; Stefano Anzellotti, Supervision, Investigation, Visualization; Michael A McDannald, Conceptualization, Resources, Data curation, Software, Formal analysis, Supervision, Funding acquisition, Validation, Investigation, Visualization, Methodology, Writing – original draft, Project administration, Writing – review and editing

### Author ORCIDs
Amanda Chu ⓘ http://orcid.org/0000-0002-5590-3245
Michael A McDannald ⓘ https://orcid.org/0000-0001-8525-1260

### Ethics
All protocols were approved by the Boston College Animal Care and Use Committee and all experiments were carried out in accordance with the NIH guidelines regarding the care and use of rats for

experimental procedures. The Boston College experimental protocol supporting these procedures is 2024-001.

### Decision letter and Author response

Decision letter https://doi.org/10.7554/eLife.82497.sa1
Author response https://doi.org/10.7554/eLife.82497.sa2

## Additional files

### Supplementary files

• Supplementary file 1. Experiment 1 ANOVA results for suppression ratio. Significant main effects and interactions are bolded.

• Supplementary file 2. Behavior definitions. Definitions are provided for each behavior scored.

• Supplementary file 3. Experiment 3 ANOVA results for suppression ratio. Significant main effects and interactions are bolded.

• MDAR checklist

### Data availability

Raw images and observer judgments are freely available: https://doi.org/10.7910/DVN/HKMUUN.

The following dataset was generated:

| Author(s) | Year | Dataset title | Dataset URL | Database and Identifier |
|---|---|---|---|---|
| Chu A, Michel CB, Gordon NT, Hanrahan KE, DuBois AM, Williams DC, McDannald MA | 2022 | Behavior frames and observer judgments from 24 rats (12 females) receiving Pavlovian fear discrimination | https://doi.org/10.7910/DVN/HKMUUN | Harvard Dataverse, 10.7910/DVN/HKMUUN |

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
