## [Editor Report]

This is an important and timely characterization of a diversity of behaviors male and female rats exhibit during the acquisition of Pavlovian fear conditioning in a conditioned suppression procedure. The data are compelling and provide an exhaustive analysis of behavior in a complex associative learning paradigm that blends aversive Pavlovian and appetitive instrumental elements. The generalizability of these findings to other paradigms could be enhanced, however, with the inclusion of tests of cue responses in a neutral environment. These findings are likely to be of interest to those who study fear conditioning and associative learning more broadly in rodents.

---

## [Decision Letter]

**Decision letter after peer review:**

Thank you for submitting your article "A fear conditioned cue orchestrates a suite of behaviors" for consideration by *eLife*. Your article has been reviewed by 3 peer reviewers, and the evaluation has been overseen by a Reviewing Editor and Kate Wassum as the Senior Editor. The following individual involved in the review of your submission has agreed to reveal their identity: Stephen Maren (Reviewer #2).

Essential revisions (for the authors):

The reviewers all agree that the inclusion of an off-baseline test would be a necessary addition for this manuscript and would really increase the scope and impact of the paper (the other cues would be interesting to test in this way, as well, but the fully reinforced danger cue seems to produce the most counterintuitive result). In the discussion, the authors attempt to make apples-to-oranges comparisons with far simpler associative learning paradigms. Partially replicating those paradigms with an additional off-baseline test in a neutral environment conducted after the differential conditioning/instrumental suppression procedure would give great insight into how these data relate to those collected by other groups. The good thing about the inclusion of this experiment is that it is interesting no matter how it works out. If there are lots of locomotor reactions to the danger cue in the off-baseline context, that would suggest overall learning history is key, whereas if the animals snap into a strategy dominated by freezing it would suggest that the associative load of the testing context plays the crucial role.

---

## [Author Response]

Essential revisions (for the authors):The reviewers all agree that the inclusion of an off-baseline test would be a necessary addition for this manuscript and would really increase the scope and impact of the paper (the other cues would be interesting to test in this way, as well, but the fully reinforced danger cue seems to produce the most counterintuitive result). In the discussion, the authors attempt to make apples-to-oranges comparisons with far simpler associative learning paradigms. Partially replicating those paradigms with an additional off-baseline test in a neutral environment conducted after the differential conditioning/instrumental suppression procedure would give great insight into how these data relate to those collected by other groups. The good thing about the inclusion of this experiment is that it is interesting no matter how it works out. If there are lots of locomotor reactions to the danger cue in the off-baseline context, that would suggest overall learning history is key, whereas if the animals snap into a strategy dominated by freezing it would suggest that the associative load of the testing context plays the crucial role.

We have performed the requested experiment. The results are described in full in the revised manuscript. In short, testing in extinction found a smaller number of danger-elicited behaviors. However, the most prominent behavior was locomotion that peaked when foot shock would have occurred